# Neck musculoskeletal model generation through anthropometric scaling

**Paulien E. Roos[1], Anita Vasavada[2], Liying Zheng[3], Xianlian Zhou[4]***

**1** Biomedical and Life Sciences Division, CFD Research Corporation, Huntsville, AL, United States of America, **2** Voiland School of Chemical Engineering and Bioengineering, Department of Integrative Physiology and Neuroscience, Washington State University, Pullman, WA, United States of America, **3** Health Effects Laboratory Division, National Institute for Occupational Safety and Health, Morgantown, WV, United States of America, **4** Department of Biomedical Engineering, New Jersey Institute of Technology, Newark, New Jersey, United States of America

* alexzhou@njit.edu

**Data Availability Statement:** All relevant data are within the manuscript and its Supporting Information files.

**Funding:** XZ and PR received funding provided by the US Army under grant W81XWH-14-C-003 for

## Abstract

A new methodology was developed to quickly generate whole body models with detailed neck musculoskeletal architecture that are properly scaled in terms of anthropometry and muscle strength. This method was implemented in an anthropometric model generation software that allows users to interactively generate any new male or female musculoskeletal models with adjustment of anthropometric parameters (such as height, weight, neck circumference, and neck length) without the need of subject-specific motion capture or medical images. 50th percentile male and female models were developed based on the 2012 US Army Anthropometric Survey (ANSUR II) database and optimized with a novel bilevel optimization method to have strengths comparable to experimentally measured values in the literature. Other percentile models (ranging from the 1st to 99th percentile) were generated based on anthropometric scaling of the 50th percentile models and compared. The resultant models are reasonably accurate in terms of both musculoskeletal geometry and neck strength, demonstrating the effectiveness of the developed methodology for interactive neck model generation with anthropometric scaling.

## Introduction

Neck pain or injury is a common issue affecting a large percentage of the population in both civilian [1] and military populations [2]. For military personnel, head supported mass (HSM) such as helmet and helmet mounted gears pose additional risks of neck injuries. For civilians, sports helmets, motorcycle helmets, or occupational head protection (e.g. construction workers' and welders' helmets) pose similar risks of neck injuries, especially due to prolonged wear. Heavy or off-balance HSM requires stronger muscle contraction to stabilize the head during different motions, which in turn increases loading to tissues of the cervical spine. Insights into neck muscle contraction and loading of the cervical spine *in vivo* are important to understand and minimize risks of chronic injury.

this study. XZ is also supported by NJIT's new faculty startup fund.

**Competing interests:** The authors have declared that no competing interests exist.

Because loading of the cervical spine cannot easily be measured *in-vivo*, modelling approaches are often used to provide estimates. For example, to estimate cervical disc compressive forces, one must consider the muscle forces acting along the cervical spine, the weight of the head and head worn mass. Several musculoskeletal models of the cervical spine have been previously developed and can be used for such estimates. Van der Horst et al. [3] developed a combined multi-body and finite element model (based on [4]) with ligaments, simplified muscles, and nonlinear stiffness of intervertebral discs. Another detailed model [5], based on imaging and cadaver dissection data [6], includes overall ligament actions, but no individual ligaments. Vasavada et al. [7] developed an advanced model with detailed muscle architecture based on cadaver dissections and refined it with accurate muscle volumes based on MRI studies [8]. A unique female neck model has been developed [9] based on the anatomical data of the Visible Human Female (VHF). This VHF neck model was developed to represent the geometry and muscles around the female head and neck. However, this model was based on a single female who happened to be obese, and the process of creating subject-specific models is still time consuming and labor intensive. These models developed by Vasavada's group do not have mass or inertia properties so they are not ready for dynamic simulations. Cazzola et al. [10] improved Vasavada's model with inertia properties and integrated it with a whole body model for rugby simulations. They also increased the isometric strength of each muscle in Vasavada's model by at least 40%. Their resultant model has a good agreement in extension strength but is still weak in flexion strength. More recently, Mortensen et al. [11] improved the Vasavada neck model with inclusion of passive elements and additional hyoid muscles. The strength of the extension muscles was further scaled by 1.4 and the flexion muscles by 2.7 in order to match experimentally measured flexion and extension neck strengths. However, this scaling also resulted in unrealistically strong hyoid muscles that produced a jaw force that is more than three times the measured value. Therefore, it remains a challenge to obtain a neck musculoskeletal model that has both realistic muscle strengths and realistic overall neck strengths.

Most existing neck models represent either a subject or a typical population and scaling these models requires either motion capture or medical image data. Desantis Klinich et al. [12] predicted cervical spine geometry based on age, height, and gender based on lateral-view radiographs of 180 adult subjects, but only in the 2D sagittal plane. It is not an easy task to scale a detailed neck musculoskeletal model to specific neck and head anthropometry (e.g. by given measured head and neck circumferences). Considering most existing neck models do not incorporate the whole body skeleton, it is even harder to scale the model with whole body anthropometry such as height and weight. In addition, existing neck scaling methods change the neck musculoskeletal geometry, individual muscle paths and forces, often without putting limits on the alteration of the overall neck strength. To predict cervical loadings accurately in dynamic simulations, model strength re-calibration is desired for subject-specific models, which is again a non-trivial task.

To address these challenges, the aim of this study was to develop methodology to quickly create anthropometric whole body models with detailed neck musculoskeletal architectures and appropriate neck strengths based on just a few whole body and neck anthropometry measurements, such as height, weight, neck circumference, and neck length. First, a male and a female 50th percentile model with detailed neck muscles were optimized to have mean neck strength (moment generation capacity). Based on user specified anthropometry parameters and the ANSUR II 3D database (an anthropometry database including both traditional measurements and 3D body scans of thousands of military personnel, including both male and female) [13], these models can be interactively scaled, which includes the scaling of the joint skeleton, mass and inertia, muscles, and strength. This allows the generation of personalized

neck musculoskeletal models with realistic strength and anthropometry that are required to address research questions that consider the effect of body size and gender on predicting *in vivo* neck loadings.

## Methods

The overall anthropometric model generation methodology consists of the following steps:

1. an existing (original) neck model was scaled and fitted to the anthropometry of the ANSUR II 50th percentile male (and female) and the segment inertia properties were calculated based on volumetric body segmentation of a 3D body;

2. maximum isometric forces of all muscles were optimized such that the overall neck strengths (in flexion, extension, lateral bending, and axial rotation) of the 50th percentile male (and female) models were close to the experimentally measured mean values (from literature);

3. lastly the 50th percentile male and female models were loaded into the Anthropometric Model Generation (AMG) software [14] and interactively scaled to generate arbitrary anthropometric musculoskeletal models. The AMG software is in-house developed software (by CFD Research) that generates anthropometric models using Principal Component Analysis (PCA).

### Scaling of original neck model to 50th percentile male (and female) anthropometry

The original model was based on the initial musculoskeletal neck model developed by Vasavada et al. [7]. The initial model, which represents an approximate 50th percentile male, has been continuously improved with new information from scientific experiments and radiographic studies [9,15]. The model components include skeletal geometry, joint kinematics, and muscles (Fig 1). This model's bones are positioned to represent the upright neutral posture based on one approximate 50th percentile individual from radiographic studies. It has 8 joints (OC-C1, C1-C2, . . ., C7-T1, OC: Occipital Condyle, C1: 1st cervical vertebra, T1: 1st thoracic vertebra), 24 degrees of freedom (DOF) and 84 muscle fascicles. The intervertebral kinematics in the neck model are prescribed as a set percentage of the overall neck angle (angle of the head relative to T1). Each intervertebral joint contributes a certain percentage to the overall

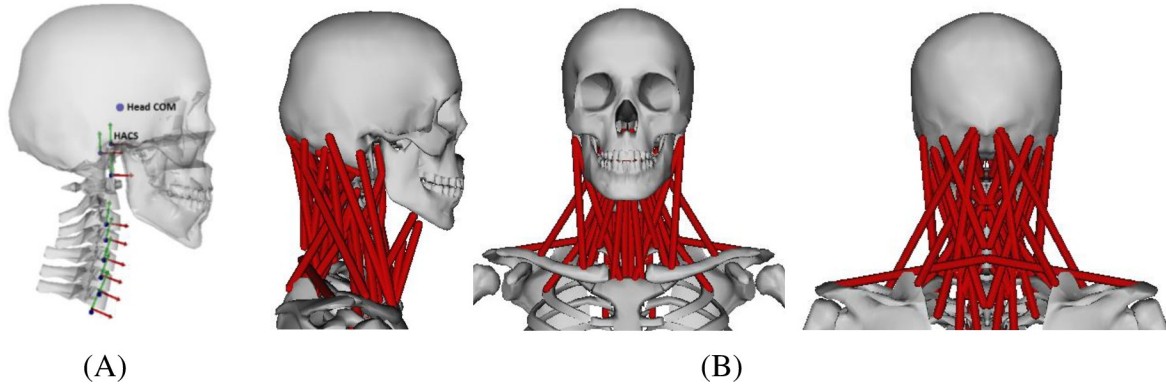

(A)                                    (B)

**Fig 1. The original neck musculoskeletal model.** (A) Skeletal joints are shown as axes and the head COM is shown as a sphere; (B) different views of the 84 neck muscles including the hyoid muscles.

angle, and this percentage is constant over the full range of motion. Muscle force-generating parameters were defined based on detailed anatomical studies of Kamibayashi and Richmond [16] and Anderson et al. [15] and revised according to the data presented by Chancey et al. [6] and Oi et al. [5]. The segment mass and inertia properties of the cervical spine and head were adapted from the literature [3,4]. To be able to simulate whole body motions, the detailed neck model was assembled onto a whole body model [17] with 61 DOFs in total.

### 50th percentile anthropometric models

The ANSUR II database [13] contains 4,802 and 1,986 3D body scans of male and female subjects respectively. After each scan was mapped with a template mesh (male or female) through deformation, all body surfaces share the same mesh triangulation (with the same number of vertices and connectivity but different vertex positions). Consequently, a mean male (and female) 3D model averaged from these templated 3D body scans can be obtained. In addition, 3D principal component analysis (PCA) with these 3D body scans can be carried out. The deviation of each model from the mean model can be calculated by subtraction. By computing the covariance matrix of all deviation vectors, one can conduct PCA and get all principal values and vectors (i.e., eigenvalues or eigenvectors). By varying the principal component (PC) weights within reasonable ranges, new synthesized 3D surface models can be easily generated. Nonetheless, these PC weights are not directly correlated to the traditional anthropometric features or measurements. For example, the first male PC correlates to the overall size change and affects many anthropometric features such as height, weight, chest breath, and torso height. We demonstrated that the method of feature analysis in the AMG software can generate any 3D body model with user specified features (measurements such as height and weight) [14].

Here we use the averaged male and female models as the corresponding 50th percentile models. The 50th percentile ANSUR II male has a height of 1.76 m, a mass of 84.6 kg, a neck circumference (at Adam's apple height) of 39.5 cm, and a neck length (defined as the vertical distance between the C7 and tragion) of 10.8 cm. The 50th percentile female has a height of 1.63 m, a mass of 66.8 kg, a neck circumference of 32.8 cm, and a neck length of 10.6 cm.

The ANSUR II dataset includes 42 body landmarks, including multiple markers on the head. In addition to these ANSUR II landmarks, 110 landmarks were identified to calculate joint center locations, body segment rotations, and additional body measurements [14]. Twelve of the ANSUR II and eight of the additional landmarks are located on the head and neck. The 3D coordinates of each landmark only need to be recorded once for the mean surface model and they can be automatically translated to other models synthesized using the AMG software due to the underlying Principal Component Analysis (PCA) data [14]. The lower neck joint center is calculated as a weighted average of the C7 and Clavicle landmarks. The skull-neck joint is located at the top of the neck between the C1 vertebra and the skull and calculated based on another two markers located on the left and right side of the head near the tragion.

To create a 50th percentile male (and female) model based on ANSUR II, the original neck model was first manually scaled and fitted inside the mean 3D body (Fig 2) and its segment inertia properties were updated based on a volumetric body segmentation of the 3D body. The manual scaling was done by scaling separate body segments to obtain a visually close-fitting match. The resulting mean model was voxelized for body segmentation based on the musculoskeletal segment definitions (Fig 3). For simplicity, the neck is segmented as a whole instead of 7 smaller cervical segments defined in the musculoskeletal model (C1-C7). Uniform density was assumed for all segments and the overall density was adjusted for the male and female separately to match the total body mass of the mean ANSUR II male and female [14]. The selected

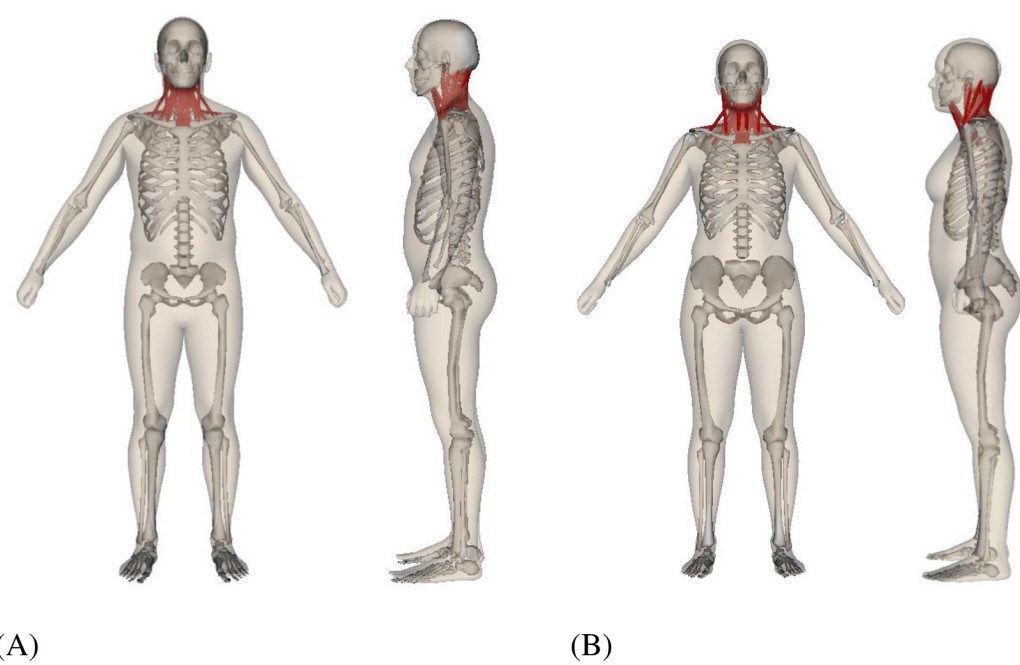

(A) (B)

**Fig 2. The adjusted 50th percentile musculoskeletal model fitted within the mean ANSUR II Skin.** (A) Male; (B) Female.

voxel volumes for the head and neck (Fig 4) were used to calculate the head and neck volume, mass, center of mass (COM), and moment of inertia (MOI). With a given density, the mass and COM of a segment can be easily computed from the sum of voxel mass and position, and the MOI can be computed with regard to the COM frame by summing over all voxels with the parallel axis theorem.

Although a female neck model has already been developed by Zheng [9], this model was based on a single female subject who happened to be obese and no mass and inertia properties were provided. The vertebral geometry and muscle attachments in this model were specific for that particular female. This makes it difficult to define differences in neck behavior between males and females using the current male and female neck models. In the studies by Zheng et al. [8] and Zheng [9], it was found that females have 59% lower neck total muscle volume (TMV) compared to males (females: $510\pm43cm^3$, males: $814\pm64cm^3$; p<0.001). However, the same authors also showed that there is no significant gender difference in vertebral shape (wedging or concavity) or in kinematic parameters such as intervertebral motion distribution or instantaneous axis of rotation when normalized by vertebral size; moreover, the muscle volume distribution is similar between males and females. Therefore, for consistency, the female musculoskeletal model was generated by manually scaling the male model while incorporating gender specific differences, such as differences in mass and inertia distributions, as well as muscle strength with regards to anthropometry.

## Optimization of maximum isometric muscle force

After geometric and anthropometric fitting of the male and female models to 50th percentile ANSUR II data, their strengths needed to be optimized to 50th percentile male and female strength. Neck strength data in literature usually report either the forces measured at certain locations on the head or the estimated moments. In most studies, the force was applied or

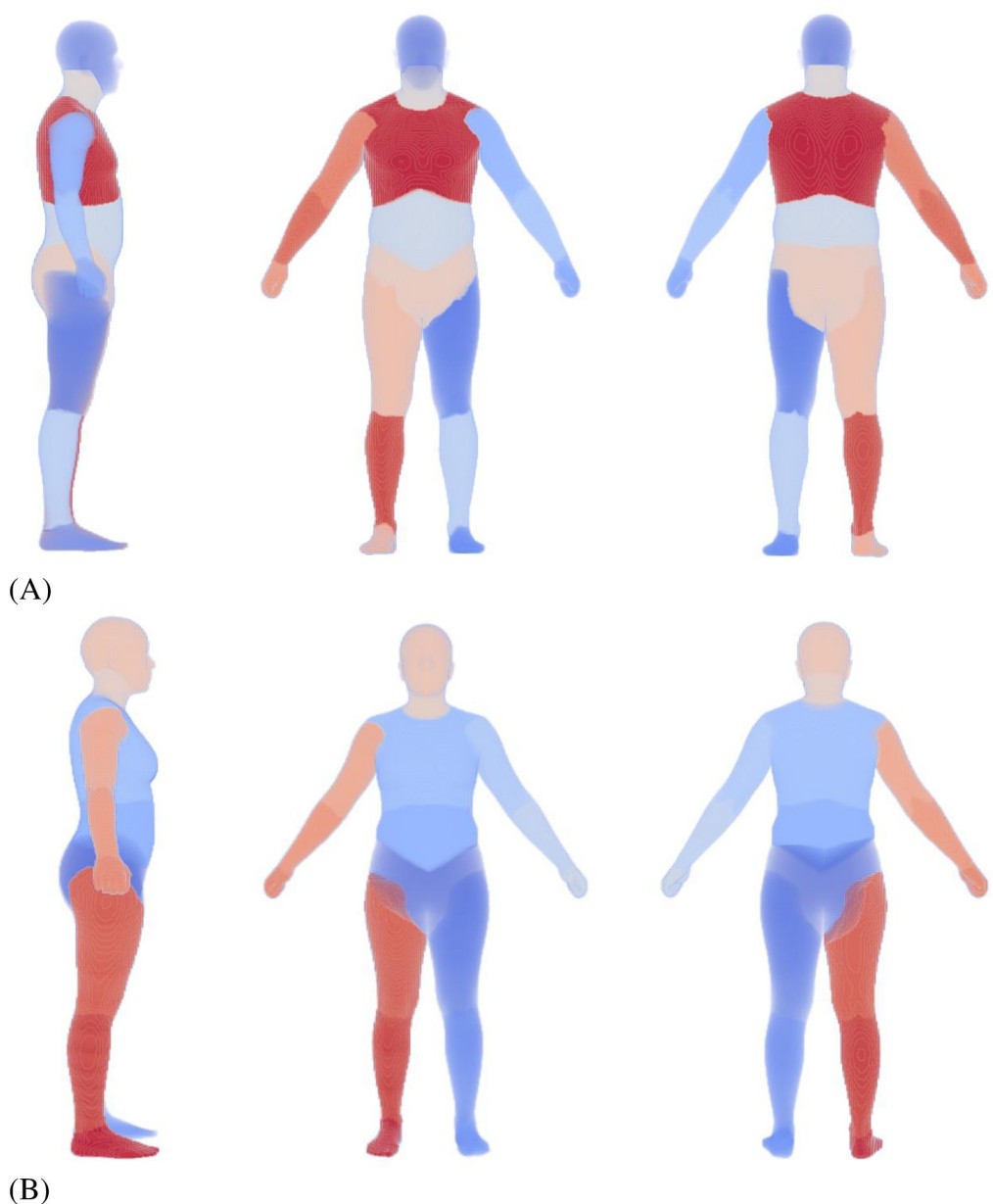

**Fig 3.** Body segmentation of the ANSUR II average (A) male and (B) female. The segmentation is done based on the anatomical structures contained in each body part and limited manual adjustment. The different shades of red and blue identify the different body parts.

measured at the forehead for flexion, at the opisthocranion for extension, and at the temple for lateral bending. There is a large variation in the strength data from literature and only [19–22] presented data for both males and females. Male flexor strength, for example, ranges from 72 to 197 N and female flexor strength from 41 to 91 N. The reported strength ratios between flexion and extension range from 58% to 85% for male and 57% to 71% for female; and ratios of female to male strength range from 0.42 to 0.68 for flexion and 0.4 to 0.74 for extension. The large variation in strength measurements made it difficult to use the averages of these studies as the target strengths of 50th percentile males and females, as this would result in different

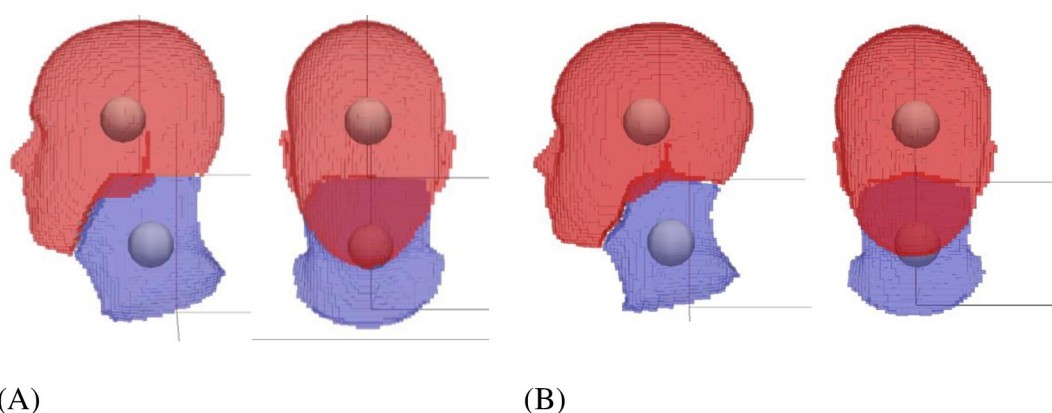

(A)                                                              (B)

**Fig 4.** Zoom-in view of the head and neck segmentation for (A) male and (B) female. The balls are the COMs of the head and neck. The local rotational axes for head and neck are also shown. Planes of separation of the neck from the torso and the head are based on the rules by Walker et al. [18].

strength ratios (between flexion and extension) for the male and females. However, other sources in literature suggested that muscle volume distribution does not differ (or only minimally) between males and females [8], implying that strength ratios shall be similar for male and female. We therefore used the male data from literature that were close to 50th percentile male and scaled the male target data for the average female with a female to male ratio of 0.65 from [21,22] since these studies measured strength in a similar way as it was calculated during our model strength optimization (explained below). This average male strength and female strength (as a ratio of male strength) that was used as a target in our optimizations is presented in Table 1.

Optimization of muscles forces requires the knowledge of physiological force limits for each muscle, which cannot directly be measured *in vivo*. Muscle force is proportional to physiological cross-section area (PCSA), with a proportionality constant known as the specific tension. PCSA is directly proportional to muscle volume and inversely proportional to fiber length, both of which can be measured with MRI or in cadavers. To ensure that the muscle force ranges remained physiologically realistic during muscle strength optimization, it was ensured that the muscle volume distribution stayed within limits reported in literature. Zheng et al. [8,9] described a muscle volume distribution and defined regression equations for total muscle volume on this same dataset (Table 2). Optimization of muscle parameters in our model may deviate from the measured percentage muscle volume distribution. We therefore constrained this optimization (for both the male and female model) to produce muscle volume

**Table 1. Peak forces and moments that can be resisted by the 50th percentile male and female for extension, flexion, lateral bending and axial rotation.** The male and female target data from literature [19–22] are the targets used for optimization, and the male and female optimized are the forces and moment that can be resisted by the optimized models.

|  | Extension (N) | Flexion (N) | Lateral bending (N) | Axial rotation (Nm) |
|---|---|---|---|---|
| Original model | 255 | 66 | 109 | 7.4 |
| Male target literature | 254 | 122 | 173 | 11.2 |
| Male optimized | 248 | 119 | 190 | 11.5 |
| Difference (%) | -2.4 | -2.5 | 9.8 | 2.7 |
| Female target literature | 165 | 79.6 | 112.3 | 7.3 |
| Female optimized | 171 | 78 | 103 | 7.3 |
| Difference (%) | 4.0 | -2 | -8.3 | 0 |

**Table 2. Muscle volume distributions (calculated from isometric muscle strength and optimal fiber length) by Zheng et al. [8,9], with total muscle volume in the bottom row.**

| | Muscle volume distribution by [8] | | |
|---|---|---|---|
| | Females (n = 3) | Males (n = 7) | Average (n = 10) |
| Sternocleidomastoid | 17.3% | 14.0% | 15.0% |
| Scalenus | 6.4% | 6.3% | 6.3% |
| Longus capitis | 2.4% | 1.6% | 1.8% |
| Longus colli | 1.9% | 1.7% | 1.7% |
| Trapezius | 25.9% | 28.7% | 27.9% |
| Splenius (capitis and cervicis) | 9.3% | 9.9% | 9.7% |
| Semispinalis capitis | 10.6% | 10.7% | 10.7% |
| Semispinalis cervicis and multifidus | 8.7% | 7.0% | 7.5% |
| Longissimus capitis | 1.5% | 1.8% | 1.7% |
| Longissimus cervicis | 1.1% | 1.3% | 1.2% |
| Levator scapulae | 8.9% | 10.0% | 9.7% |
| Rectus capitis major | 0.8% | 0.8% | 0.8% |
| Rectus capitis minor | 0.3% | 1.1% | 0.9% |
| Obliqus capitis superior | 0.3% | 0.6% | 0.5% |
| Obliqus capitis inferior | 1.4% | 1.9% | 1.8% |
| Infrahyoids | 3.4% | 2.7% | 2.9% |
| Total neck muscle volume (cm$^3$) | 510.4±43.0 | 813.9±63.6 | |

distributions that are within ±5% of those reported by Zheng et al. [8,9]. Zheng's muscle volume percentage distribution data was used because they are all based on the same subjects and scans of living subjects instead of cadaver measurements.

The muscle strength of the original neck model (see section 1. Scaling of original neck model to 50th percentile male (and female) anthropometry) was based on muscle PCSA and fiber length measurements on cadavers [16]. The overall neck strength was computed by simulating the experimental conditions with increasing forces or (axial rotation) moment applied to the model until it can no longer maintain the static neutral posture. For example, to determine the flexion strength, we applied an increasing force on the forehead and conducted static optimization to determine if the muscles can coordinate to generate the required joint torques. The computed overall neck strength (moment generation capacity) of the original or scaled 50th percentile neck model did not agree well with that reported in literature for a 50th percentile male (Table 1). The model was too strong for neck extension and too weak for neck flexion. Since most muscles have multiple functions (such as extensor and axial rotator), the model cannot easily be manually tuned or scaled to agree better with experimental data in all directions. For this reason, we developed an optimization routine that matches the moment generating capacity of the model to average moment generating capacity data of 50th percentile males measured experimentally (target neck strength as listed in Table 1). This optimization routine can vary the peak isometric force of all or selected muscles between minimum and maximum values reported in literature [5,7,9,23,24], or any other predefined range. This ensures that all muscle parameters stay within their reasonable physiological ranges.

For strength optimization, we used outcomes from studies that reported forces at the head except for the axial rotational moment. Only maximum isometric forces of muscles were optimized such that the model could resist the maximum force applied that corresponded to the values from literature and the ability to resist higher forces was penalized in the optimization formulation. The location of the point of force application to the skull was defined based on

the anatomical landmarks on our mean 3D male and female skin models. The forces (or moment for rotation) and location of application are presented in Table 1 for the male and female model.

The objective function used in the optimization is as follows:

$$J = \sum_{k=1}^{N=4} w_k \left( \sum_{i=1}^{n} \left( \tau_i^{sim} - \tau_i^{exp} \right)^2 \right)$$

with $\tau_i^{exp}$ as the target joint torques (at all cervical joints) required to resist the experimentally measured forces from literature, and $\tau_i^{sim}$ as the maximum attainable joint torques from the muscles for a given set of muscle parameters. $n$ is the number of cervical joints. $N = 4$ indicates the four experiment modes included (flexion, extension, bending and rotation). In our optimization, we used equal weights, $w_k = 1.0$ for all modes. The joint torques $\tau_i^{sim}$ can be obtained through a separate inner static muscle optimization, which optimizes all muscle forces in order to produce the target torques $\tau_i^{exp}$. The objective function of this inner static optimization is

$$J_s = \sum_{i=1}^{m} \left( \frac{f_i}{f_i^{max}} \right)^2 + w_\tau \sum_{i=1}^{n} \left( \tau_i^{sim} - \tau_i^{exp} \right)^2$$

in which $m$ is the number of muscles, $f_i$ are the muscle forces to be optimized, $f_i^{max}$ is the maximum isometric muscle forces, $\tau_i^{sim}$ are the joint torques generated by all muscle forces, $w_\tau$ is a large weight (for which we used 100 in our optimization). The first term aims to minimize the muscle effort or activation and the second term aims to minimize $\tau_i^{exp} - \tau_i^{sim}$, often referred as the residual torques in a muscle static optimization problem. The computed residual torques also appear in the outer objective function $J$ above.

However, use of the objective function $J$ in strength optimization will likely produce a strong model with unnecessary high strength because it can easily generate the required $\tau_i^{sim}$ to be equal or close to $\tau_i^{exp}$ even with sub-maximum muscle forces to minimize $J$. To determine if this happens, we artificially increase the experiment forces by a small ratio (e.g. 2%) that changed $\tau_i^{exp}$ to $\tau_i^{exp'}$ and redo the static muscle optimization. If the resultant $J$ is smaller than a tolerance (e.g. 1e-3), it means the current muscles can generate torques more than necessary and are too strong. Therefore, we added an additional penalty term $J_p$ to the objective function $J$ above such that the final objective function $J_F = J + J_p$ with

$$J_p = \sum_{k=1}^{N=4} w_k \frac{1}{\left( \sum_{i=1}^{n} \left( \tau_i^{sim} - \tau_i^{exp'} \right)^2 \right)}.$$

The hyoid muscle groups in the original neck model were included in the optimization, assuming they participate in maximum voluntary contraction experiments, where the jaw could be clenched with force contributions from hyoid muscles.

The overall optimization method is a bilevel optimization process. On the top, a global optimizer was used to search the entire parameter space for optimal parameters that minimize the objective functions above. While evaluating the objective functions, the global optimization involves an inner static muscle optimization that predicts $\tau_i^{sim}$. Typically, there is no guarantee the optimization outcome is the global optimum since the objective never reaches zero (satisfying the strength objective for all four modes) within limited time (e.g. a half hour).

To prevent some individual muscle volumes from becoming too large within a muscle group, individual muscle volume percentages within each group were approximately maintained within the optimization. Since Zheng et al. [8] only presented muscle volumes for

muscle groups instead of the individual muscles, the total volume of each muscle group was distributed over individual muscles based on their proportion in the original male neck model. Therefore, the muscle group volume distribution is based on Zheng et al., while the ratio of each individual muscle subvolume within a group (such as sterno-mastoid, cleido-mastoid and cleido-occipital for the sterno cleido mastoid muscle group) was based on the original model. During the optimization, the left-right symmetry of all muscles was enforced through equality constraints of muscle parameters.

## Anthropometric scaling of musculoskeletal models

To create body surface models of different anthropometry, the AMG software [14] uses virtual body measurements, such as segment lengths, width, depths and circumferences, calculated from digital body landmarks on the 3D body. It links traditional 1D anthropometry measurements with 3D principal components and allows users to directly change anthropometry parameters to manipulate the body shapes and vary inertia properties accordingly (Fig 5).

As mentioned earlier, during anthropometric model generation, the new joint locations are determined by the positions of surface landmarks and the new mass and inertia properties of each segment are determined by the voxelized segmentation. By linking the body-surface-model-determined joint locations and inertia with the musculoskeletal model, the AMG software can interactively scale the musculoskeletal model simultaneously with the 3D surface model. Scaling of the neck segment is based on the neck circumference at the Adam's apple height and the total neck segment length. The scaling factor is computed by comparing the values of the current model with those of the mean ANSUR II male or female model. The 3D segmented model has one single neck segment instead of 7 cervical spine segments. Therefore, we scaled these cervical segments based on their geometry and mass distribution in the original model. The 3D model has no guaranteed symmetry between left and right, but the musculoskeletal model can be symmetrized using the average of the left and right values when needed (e.g. during output).

Fig 6 shows examples of 5th, 50th and 95th percentile male and female anthropometry models with specified height, weight, neck circumference, and neck length (Table 5 and Table 6). The values of these features can then be adjusted, and the body shape will change accordingly.

The geometrical and physical parameters of each muscle were also scaled based on neck anthropometry. During the anthropometric scaling, the position of each muscle path point (called node here) is scaled with its attached segment (which is scaled in XYZ directions with

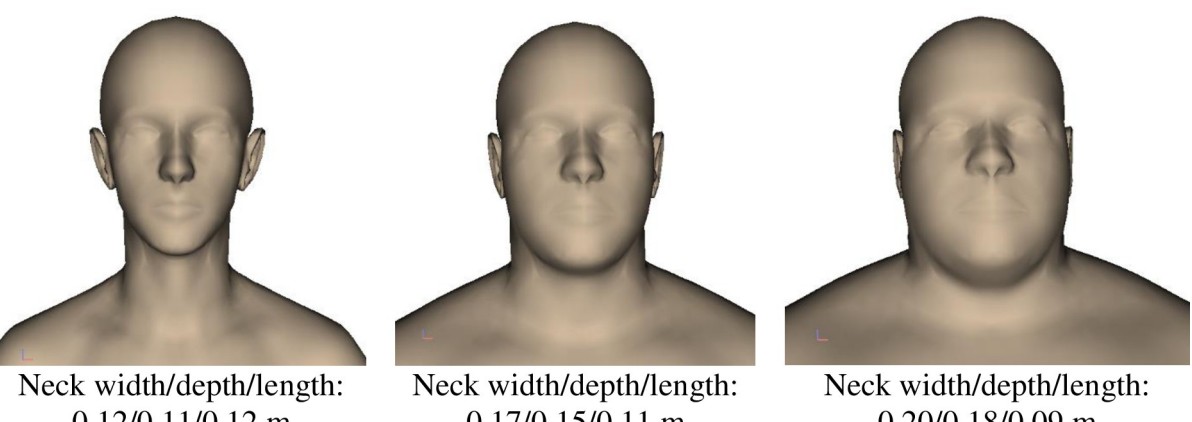

Neck width/depth/length:
0.12/0.11/0.12 m

Neck width/depth/length:
0.17/0.15/0.11 m

Neck width/depth/length:
0.20/0.18/0.09 m

**Fig 5. Exemplary anthropometry body models generated based on neck width, depth and length.**

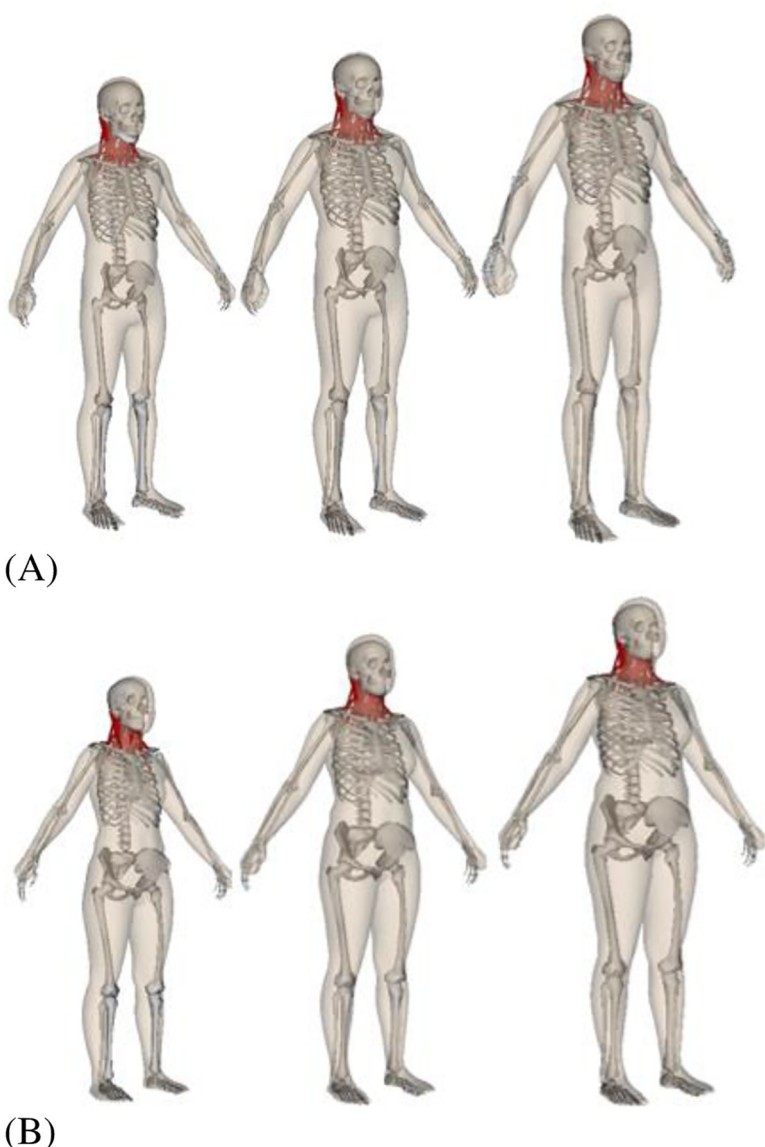

**Fig 6.** Anthropometrically scaled percentile models (5th, 50th, 95th) for (A) males and (B) females.

factors $s_x$, $s_y$, and $s_z$. This geometry scaling causes the muscle to change its path line and its total length changes from $L_0$ to $L$. The muscle fiber length and tendon slack length is scaled by the muscle length scaling factor ($s_L = L/L_o$), similar to the scaling law employed in [25,26]. For the max muscle fiber force, we scaled it with the ratio of muscle PCSA ($s_c$) before and after the scaling. Nonetheless, it is not straightforward to derive the muscle PCSA scaling factor $s_c$ from the segment scaling factors $s_x$, $s_y$, and $s_z$ since the cross-section may not align with any of the XYZ planes. To address this problem, we assume each muscle node $i$ has a volume $v_i$ that is scaled by a scaling factor $s_{v_i} = s_{x_i} \times s_{y_i} \times s_{z_i}$. Then the total volume scaling factor for the muscle can be defined as a weighted average of the nodal volume scaling factors

$$s_v = \frac{\sum_i^n l_i \times s_{v_i}}{\sum_i^n l_i} = \frac{\sum_i^n l_i \times s_{v_i}}{L} = \sum_i^n \frac{l_i}{L} \times s_{v_i}$$

in which $n$ is the total number of nodes in this muscle, $l_i$ is the characteristic length of node $i$ (defined as the half length of the edge/s connecting this node to its neighbors), $L = \sum l_i$ is the total length of the muscle, and $\frac{l_i}{L}$ is the volume scaling weight factor for the $i^{th}$ node. With both $s_L$ and $s_v$ given above, the PCSA scaling factor can then be computed as $s_c = s_v/s_L$.

## Results

### Mass and inertia properties

Systemic methods that use geometric approximations or predefined anthropometric features (such as [27,28]) are fairly accurate in estimating body segment moments of inertia (MOI) of the upper and lower extremities but may not be accurate enough for the head and neck. Our voxelized segmentation method captures the fine details of the anthropometric body variation without approximation and offers better representation of mass and inertia properties. The calculated mass of the head and neck for the 50th percentile male and female model (Table 3), based on the volumetric segmentation in Fig 3 and Fig 4, agreed well with literature [18,29–31]. The COM of the head and neck is further forward and higher than that reported by [32], [29], and [33], even though the axes definitions are similar to that in our model. This could be because of the definition of our neck and head segments. Our definition has a slightly more detailed separation between the cervical spine and the skull. The neck COM is difficult to compare, because of the difference in the location of the axes. The head MOI (Table 3) estimated for our model is in good agreement with that from literature [18,29,30,32,33], while the neck MOI is higher than that reported by McConville et al. [32]. This could be because of the differences in the definitions of the neck segment or the measurement method.

### Neck strength optimized models

The original male model was too weak in flexion, lateral bending, and axial rotation and too strong in extension (Table 1). After optimization, the strength of the 50th percentile male was improved for most directions from a maximum of 45% to be below 3% (Table 1). However, the optimized male model was still much weaker in flexion than the experimental measured value (9.8%). Closer investigation of the optimization results showed that the male model was not capable of producing sufficient flexion strength at the top cervical vertebrae without the optimized parameters deviating too much from reasonable values. Therefore, an additional flexor muscle, the rectus capitis anterior muscle, which was not included in the original model, was added. To complete the rectus capitis muscle group, the rectus capitis lateralis was added as well. Their locations were based on anatomy of these muscles (Fig 7) and their initial strengths were based on [34] with a maximum isometric force of 32.5N. However, to consider the discrepancy in reported specific muscle tension, ranging from 35 N/cm$^2$ to 137 N/cm$^2$ in the literature [35], we allowed their strength to change up to a few times higher. To maintain agreement between the male and female model, identical muscles were added to the female model with scaled down strength.

**Table 3. Mass and inertia properties of the head and neck of the 50th percentile male and female models.** x, y, z are the anterior-posterior, media-lateral, and top-bottom directions, respectively. Inertia properties (unit: $10^{-4}$ *kgm*$^2$) are relative to the segment's COM.

| | | Mass (kg) | Ixx ($10^{-4}$ kgm$^2$) | Iyy ($10^{-4}$ kgm$^2$) | Izz ($10^{-4}$ kgm$^2$) |
|---|---|---|---|---|---|
| **Male** | **Neck** | 1.66 | 41.5 | 38.9 | 37.4 |
| | **Head** | 4.20 | 191.9 | 229.9 | 168.8 |
| **Female** | **Neck** | 1.22 | 27.3 | 25.3 | 22.4 |
| | **Head** | 3.61 | 143.3 | 182.1 | 143.0 |

**Rectus capitis anterior**

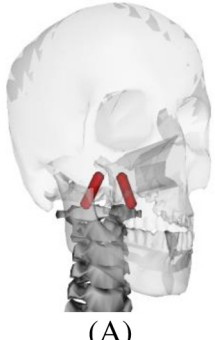

(A)

**Rectus capitis lateralis**

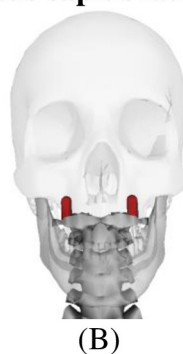

(B)

**Fig 7.** Placement of (A) the rectus capitis anterior and(B) lateralis in the model.

The final strength results of the male and female models are presented in Table 1 and the isometric forces of the individual muscles in Table 4. For both the male and female models, the strengths are very close for extension, flexion and axial rotation, all within 4%. However, the difference in lateral bending moment is relatively higher at ~8–10%.

## Percentile models

Once the optimized 50th percentile models were obtained and loaded into the AMG software, we were able to interactively generate anthropometric musculoskeletal models based on the geometrical scaling method presented in [14] and the muscle scaling method presented earlier. To demonstrate the capabilities of anthropometric scaling of these 50th percentile musculoskeletal models, 12 male and 12 female models were created based on the body height, mass, neck circumference (at Adam's apple height), and neck length specifications, corresponding to 1st to 99th percentile males and females from the ANSUR II data base ([36]; Table 5 and Table 6).

Fig 6 shows the generated anthropometric models of 5th, 50th, and 95th percentile for both female and male models. The musculoskeletal neck model is shown together with the 3D mesh model to demonstrate how they are scaled together. Fig 8 compares the updated peak forces or moments (similar to Table 1) that can be resisted by the scaled percentile models. For the male model, extensor strength increased for models with percentiles, except for the 40th and 95th percentile models whose values are slightly smaller than the models preceding them. Flexion strength increased mostly for models with percentiles, except for the 5th, 40th, and 90th percentile models. Lateral bending strength also increased mostly with percentiles, except for the 40th, 70th, and 95th percentile model. Axial rotation strength showed a very similar pattern to lateral bending, except for the 90th percentile model. For the female model, strengths increased consistently with percentile models for all moment directions.

## Discussion

The aim of this study was to develop anthropometrically scaled neck musculoskeletal models and validate their strengths. 50th percentile male and female full body musculoskeletal models with detailed neck musculature were developed and optimized. The strengths of optimal 50th percentile models are close to target values in flexion, extension, and axial rotation, all within 10% differences or less. The lateral bending strength was however relatively high in the male model (9.8%) and relatively low for the female model (-8.3%). This is likely because the female has a smaller neck circumference than the male (Table 5 and Table 6) despite similar neck

**Table 4. Maximum isometric muscle forces of the original model and the final optimized and scaled male and female models.**

| | Maximum Isometric Force (N) | | |
|---|---|---|---|
| | **Original model** | **Optimized male model (final)** | **Optimized female model (final)** |
| stern_mast | 86.1 | 221.58 | 88.4 |
| cleid_mast | 43.1 | 123.3 | 44.5 |
| cleid_occ | 43.1 | 123.3 | 43.0 |
| scalenus_ant | 65.8 | 74.07 | 65.2 |
| scalenus_med | 65.8 | 77.77 | 60.2 |
| scalenus_post | 36.8 | 43.47 | 37.3 |
| long_cap_sklc4 | 48.0 | 123.57 | 50.8 |
| long_col_c1thx | 14.4 | 41.2 | 19.2 |
| long_col_c1c5 | 14.4 | 41.2 | 20.4 |
| long_col_c5thx | 14.4 | 41.2 | 11.8 |
| trap_cl | 132.0 | 155.93 | 122.3 |
| trap_acr | 348.6 | 411.83 | 327.8 |
| splen_cap_sklc6 | 55.0 | 65.02 | 44.1 |
| splen_cap_sklthx | 53.2 | 62.84 | 48.4 |
| splen_cerv_c3thx | 50.1 | 59.16 | 29.8 |
| semi_cap_sklc5 | 91.7 | 108.3 | 74.8 |
| semi_cap_sklthx | 101.5 | 119.92 | 77.2 |
| levator_scap | 109.2 | 128.99 | 99.5 |
| longissi_cap_sklc6 | 34.3 | 40.54 | 31.0 |
| longissi_cerv_c4thx | 52.2 | 61.71 | 41.1 |
| iliocost_cerv_c5rib | 36.4 | 43.0 | 34.7 |
| rectcap_post_maj | 58.8 | 69.46 | 38.4 |
| rectcap_post_min | 32.2 | 46.55 | 29.3 |
| obl_cap_sup | 30.8 | 40.02 | 25.8 |
| obl_cap_inf | 68.3 | 80.70 | 50.5 |
| omohyoid | 26.3 | 75.2 | 26.5 |
| sternohyoid | 20.3 | 58.1 | 21.1 |
| sternothyroid | 22.8 | 65.2 | 23.4 |
| semi_cerv_c3thx | 107.1 | 126.54 | 102.5 |
| supmult-C4/5-C2 | 14.7 | 17.39 | 9.7 |
| supmult-C5/6-C2 | 19.3 | 22.77 | 17.6 |
| supmult-C6/7-C2 | 15.8 | 18.71 | 14.0 |
| supmult-T1-C4 | 16.3 | 19.28 | 10.8 |
| supmult-T1-C5 | 11.7 | 13.80 | 8.8 |
| supmult-T2-C6 | 6.5 | 7.65 | 3.9 |
| deepmult-C4/5-C2 | 7.4 | 8.79 | 4.4 |
| deepmult-C5/6-C3 | 12.3 | 14.55 | 9.6 |
| deepmult-C6/7-C4 | 16.1 | 18.99 | 9.5 |
| deepmult-T1-C5 | 12.3 | 14.55 | 7.5 |
| deepmult-T1-C6 | 8.3 | 9.83 | 7.7 |
| deepmult-T2-C7 | 14.0 | 16.54 | 9.1 |
| deepmult-T2-T1 | 14.0 | 16.54 | 9.1 |
| rectcap_ant | - | 92.6 | 60.19 |
| rectcap_lat | - | 92.6 | 60.19 |

**Table 5. Body height, mass, neck circumference and length of the 12 percentile male models that have been created together with data for the 50th percentile male.** These data are reproduced from the ANSUR II database [36].

| Male | | | | |
|---|---|---|---|---|
| Percentile | Stature (m) | Body Mass (kg) | Neck Circumference (cm) | Neck Length (cm) |
| 1st | 1.60 | 57.8 | 34.6 | 8.0 |
| 5th | 1.65 | 64.4 | 36.0 | 8.7 |
| 10th | 1.67 | 68.2 | 36.7 | 9.2 |
| 20th | 1.70 | 73.4 | 37.5 | 9.7 |
| 30th | 1.72 | 77.4 | 38.3 | 10.1 |
| 40th | 1.74 | 81.0 | 38.9 | 10.4 |
| 50th | 1.76 | 84.6 | 39.5 | 10.8 |
| 60th | 1.77 | 88.0 | 40.2 | 11.1 |
| 70th | 1.79 | 92.0 | 41.0 | 11.4 |
| 80th | 1.81 | 96.6 | 41.8 | 11.8 |
| 90th | 1.84 | 104.4 | 43.2 | 12.4 |
| 95th | 1.87 | 110.7 | 44.3 | 12.9 |
| 99th | 1.93 | 124.7 | 46.8 | 13.8 |

lengths (see also [37]). The neck circumference affects the scaling of muscle path and their moment arms, especially on bending. In addition, many muscles were modeled as straight lines, while in real life these are closer to the body and would have smaller moment arms. Muscles can be modeled as running closer to the body using wrapping objects or via points. Such a neck model has been developed by Suderman et al. [23,38,39],but they cautioned that the model can be very sensitive to wrapping object or via point kinematics and inter-individual differences in muscle paths and joint kinematics.

Several studies in literature have performed comparable muscle parameter optimization studies [40–42]. Some used Monte Carlo methods to match muscle activation during a particular movement [41,42] and compared whether muscle parameters were within physiological limits after the optimizations. Others explored the effects of measurement errors during experimental data collection and parameter estimation during inverse kinematics and dynamics

**Table 6. Body height, mass, neck circumference and length of the 12 percentile female models that have been created together with data for the 50th percentile female.** These data are reproduced from the ANSUR II database [36].

| Female | | | | |
|---|---|---|---|---|
| Percentile | Stature (m) | Body Mass (kg) | Neck Circumference (cm) | Neck Length (cm) |
| 1st | 1.48 | 46.4 | 29.1 | 7.9 |
| 5th | 1.53 | 51.3 | 30.2 | 8.7 |
| 10th | 1.55 | 54.6 | 30.7 | 9.1 |
| 20th | 1.58 | 58.5 | 31.3 | 9.6 |
| 30th | 1.60 | 61.6 | 31.9 | 10.0 |
| 40th | 1.61 | 64.5 | 32.4 | 10.3 |
| 50th | 1.63 | 66.8 | 32.8 | 10.6 |
| 60th | 1.64 | 69.5 | 33.3 | 10.9 |
| 70th | 1.66 | 72.6 | 33.8 | 11.3 |
| 80th | 1.68 | 76.4 | 34.5 | 11.6 |
| 90th | 1.71 | 82.4 | 35.5 | 12.2 |
| 95th | 1.74 | 87.1 | 36.3 | 12.7 |
| 99th | 1.78 | 98.3 | 38.5 | 13.6 |

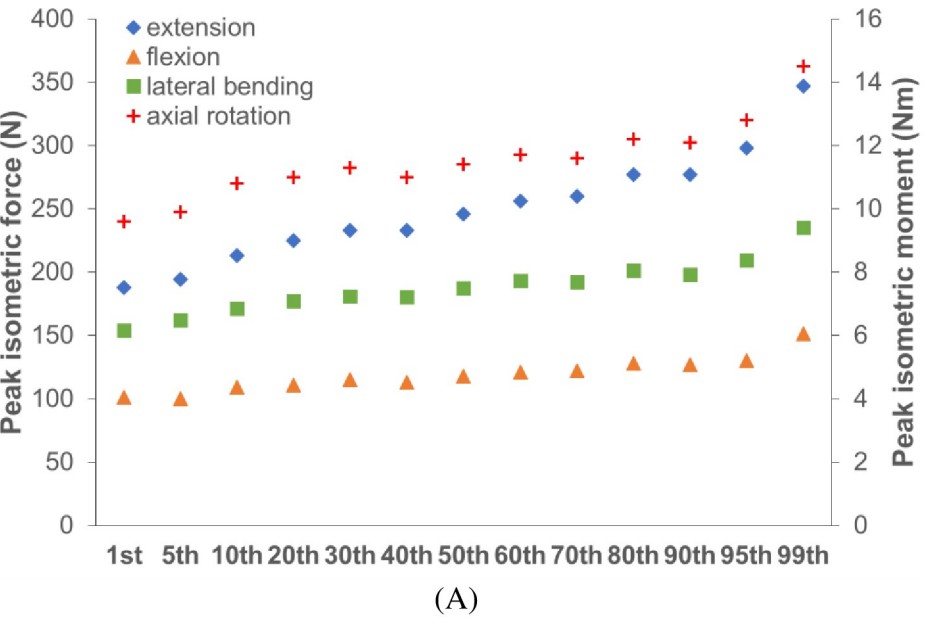

(A)

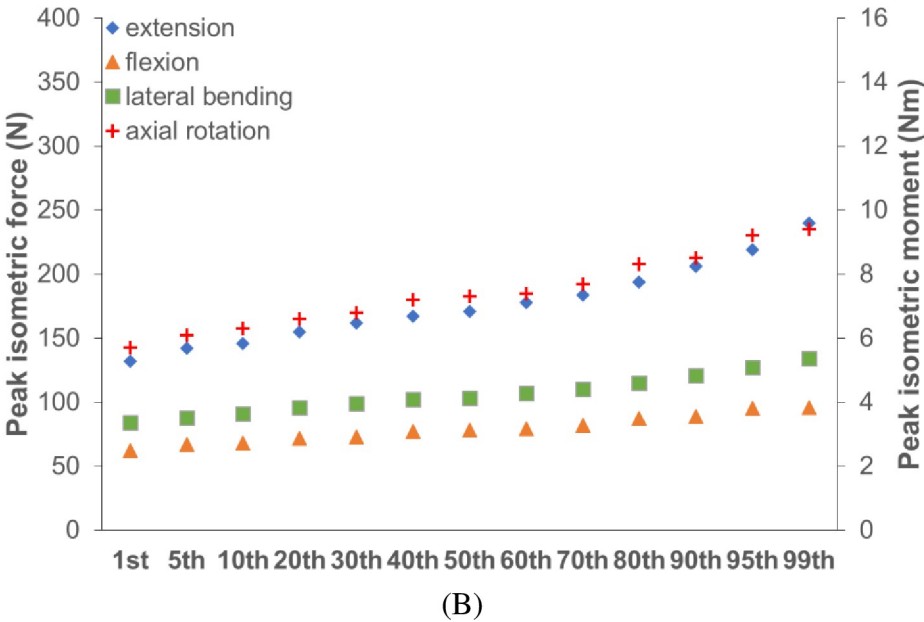

(B)

**Fig 8.** Comparison of the peak resistible forces and moments for the 13 (A) male percentile models and (B) female percentile models.

[40]. The novelty of our optimization method resides in the bilevel optimization process that employs a global optimizer for parameter sampling and a local gradient based optimizer for static muscle torque prediction. The developed optimization method has the advantages in its versatility and capability in maintaining muscle parameters automatically within physiological limits.

The strength data of the different percentile models were compared to each other. The general trend was as expected, strength increased with increasing percentile. For the male model,

there were however some instances where the next percentile had slightly lower or similar strength to the previous percentile model. This could be due to the scaling methods and non-proportional increase of neck length and circumference. In our method, muscle optimal forces were scaled with respect to estimated muscle volume and muscle paths (that determine moment arms) were scaled based on skeletal geometry. As a result, strength was not scaled linearly. While increase of neck circumference generally increases the muscle forces and moment arms, increase of neck length (or height) decrease the muscles' capability to resist forces applied at those specific application locations on the head. The two competing trends could lead to non-monotonical increases or even decreases of neck strength with percentile, if no further muscle optimization is performed.

For maximum isometric muscle force optimization, the physiological ranges of muscle forces were constrained by muscle volume distributions. It is difficult to compare muscle volume distributions between the different neck models since total muscle volumes often differ and sometimes models have different numbers of muscles included. For example, the models by Oi et al. [5] and Borst et al. [24] do not include exactly the same muscles. There is a large variation in muscle volume distribution, even in the neck models developed by the Vasavada research group. The original model had relatively low sternocleidomastoid and levator scapulae volumes, while volumes of semispinalis cervicis and multifidus, and longissimus cervicis were relatively high. The combined semispinalis cervicis and multifidus, however, deviated a lot from the muscle volumes reported by Zheng et al. [8,9]. Since there was such a large difference in neck muscle volume distribution between the different models, we also compared their PCSA distribution. The muscle volumes in the different models are based on experimental measurements and combined with measurements of fiber lengths to obtain PCSAs. The PCSA distribution was similar between the models for most muscles. The PCSAs in Oi and Pandy's model [5] were the same as those in our original model, while the PCSAs in Suderman's model [39] and Borst's model [24] were very different. Zheng's 50[th] percentile male was also very similar in PCSA to our model, only the multifidus muscles were different.

To accurately represent muscle strengths, individual muscle volume, directly related to the muscle's PCSA, or force generation capability, must be known. Zheng et al. [8] showed that individual muscle volume proportions (the ratio of the individual neck muscle volume to the total neck muscle volume) are almost fixed or insensitive to anthropometry. In addition, these volume proportions are not gender specific for most neck muscles, although small gender differences existed for three neck muscles (obliquus capitis inferior, longus capitis, and sternocleidomastoid). Based on the above findings, we can create subject-specific or percentile neck models by scaling the generic male or female model accordingly.

There are some limitations in the approach of this study. A constant body density was assumed, while in reality density of the different body segments will differ depending on their bone, fat, and muscle mass content. Furthermore, the density was chosen to match the mass of the 50[th] percentile ANSUR II male and female to the volume of their 3D model. This means that the mass of a different body composition may be slightly under or over estimated. However, for the current study, this is deemed acceptable, as body density cannot easily be predicted by anthropometry alone. Future improvement can be made by specifying body part specific mass density.

It should also be noted that an anthropometric model generated with AMG software represents the average person with user-provided anthropometric measurements. The models are not personalized at the level of vertebral geometry, which would require MRI or CT scans. In this study, percentile models were generated to represent specific percentiles from the ANSUR II database. The developed methodology can also be used to represent the anthropometry of a specific person and a larger number of measurements can be used if more details are desired.

However, as mentioned above, average body density is used, so body fat and muscle percentage is not taken into account. Also, the strength of the scaled model will be that of an average person of that anthropometry and not of the specific person. Nonetheless, strength could be further personalized with the bilevel optimization method presented here if subject specific dynamometer measurements of neck strengths are given.

## Conclusions

In conclusion, a new methodology was developed to quickly generate anthropometric neck musculoskeletal models that were interactively scaled for anthropometry and muscle strength. This method was implemented in an anthropometric model generation software that allows users to generate new musculoskeletal models with interactive adjustment of anthropometric parameters (such as height, weight, neck circumference) without the need of subject-specific motion capture or medical images. 50th percentile male and female models based on the ANSUR II database were developed and optimized with a novel bilevel optimization method to possess strengths comparable to experimentally measured values in the literature. Other percentile models generated from automated scaling of the 50th percentile models were also presented and compared. The resultant models are reasonably accurate in terms of both musculoskeletal geometry and strength, which proves the effectiveness of the developed methodology. We also applied the same methodology for anthropometric scaling of other musculoskeletal models such as upper extremity models and lumbar spine models for different applications [43]. Our method provides the capability to interactively generate accurate human musculoskeletal models with anthropometric scaling and a fast and convenient way to produce custom models for dynamic musculoskeletal simulations and analyses.

## Disclaimer

The findings and conclusions in this report are those of the authors and do not necessarily represent the official position of the National Institute for Occupational Safety and Health, Centers for Disease Control and Prevention.

## Supporting information

**S1 Data.**
(XLSX)

## Acknowledgments

We would like to thank Austin Mituniewicz and Timothy Zehnbauer for their support with generating the different percentile models.

## Author Contributions

**Conceptualization:** Paulien E. Roos, Xianlian Zhou.

**Data curation:** Xianlian Zhou.

**Formal analysis:** Paulien E. Roos, Anita Vasavada, Liying Zheng, Xianlian Zhou.

**Funding acquisition:** Xianlian Zhou.

**Investigation:** Paulien E. Roos, Xianlian Zhou.

**Methodology:** Paulien E. Roos, Xianlian Zhou.

**Project administration:** Xianlian Zhou.

**Resources:** Xianlian Zhou.

**Software:** Paulien E. Roos, Xianlian Zhou.

**Supervision:** Xianlian Zhou.

**Validation:** Paulien E. Roos, Xianlian Zhou.

**Visualization:** Paulien E. Roos, Xianlian Zhou.

**Writing – original draft:** Paulien E. Roos, Xianlian Zhou.

**Writing – review & editing:** Paulien E. Roos, Anita Vasavada, Liying Zheng, Xianlian Zhou.

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
