## [Decision Letter · Decision Letter 0]

14 Oct 2019

PONE-D-19-18973

Neck Musculoskeletal Model Generation through Anthropometric Scaling

PLOS ONE

Dear Dr. Zhou,

Thank you for submitting your manuscript to PLOS ONE. After careful consideration, we feel that it has merit but does not fully meet PLOS ONE’s publication criteria as it currently stands. Therefore, we invite you to submit a revised version of the manuscript that addresses the points raised during the review process.

We would appreciate receiving your revised manuscript by Nov 28 2019 11:59PM. To enhance the reproducibility of your results, we recommend that if applicable you deposit your laboratory protocols in protocols.io, where a protocol can be assigned its own identifier (DOI) such that it can be cited independently in the future. For instructions see: http://journals.plos.org/plosone/s/submission-guidelines#loc-laboratory-protocols

We look forward to receiving your revised manuscript.

Kind regards,

Bernadette Ann Murphy, PhD

Academic Editor

PLOS ONE

**Journal Requirements:**

**Comments to the Author**

1. Is the manuscript technically sound, and do the data support the conclusions?

Reviewer #1: Yes

Reviewer #2: Partly

2. Has the statistical analysis been performed appropriately and rigorously? 

Reviewer #1: N/A

Reviewer #2: N/A

3. Have the authors made all data underlying the findings in their manuscript fully available?

Reviewer #1: Yes

Reviewer #2: Yes

4. Is the manuscript presented in an intelligible fashion and written in standard English?

Reviewer #1: Yes

Reviewer #2: Yes

5. Review Comments to the Author

Reviewer #1: This is a very well written paper that provides a rather elegant and useful solution to generate subject-specific models of the neck. The use of a bilevel optimisation is a great idea. Also the discussion and description of the improvements with respect to previous methods is very clear and valid. However there are several points which appear less lear within the paper:

Line 67 - Please provide little more info on the ANSUR II 3D database, and briefly explain why this approach can make the difference in addressing the previously highlighted challenges.

Line 82. Please provide more info on AMG software and how it can be used. More info is provided later on (Line 124), but an initial clarification would be helpful.

Line 123. Please clarify what ‘synthesised’ mode the authors are referring to (I guess the ANSUR II model within the AMD).

Line 129. Please clarify what ‘manually’ means? Was the optimal fir chosen visually?

Fig 3. Please clarify the colour coding and its relation with density (why different tones of blue and red?)

Line 218. Please clarify the terminology used within the whole paper in relation to the different models used. Is the original neck model, the Vasavada’s (+ hyoid) model?

Line 238. Please clarify what the authors mean with a ‘inner static muscle optimisation’ as it can be mixed up with a classic static optimisation procedure (which would need to solve for the muscle redundancy problem). Therefore the authors should clarify how the muscle activations were set to generate the target joint torques (e.g. max muscle activation). Otherwise the authors should clarify how the muscle redundancy problem was solved within the bilevel optimisation (e.g. minimising sum of muscle forces?). How did the optimiser ensured a physiological muscles activation?

Also, how did the authors ensured symmetry between the right and left side of the model? The optimiser might find asymmetrical solutions. If this is dealt with as described in Lines 298-300, it should be better clarified beforehand.

The authors should move the description fo the optimisation methods (Line 262-268) before the description of the objective function.

Line 250. Please clarify what the final objective function formula (J + Jp ?)

Line 244-250. Unfortunately this explanation is not clear and I require the authors to provide the optimisation results, and a breakdown of the optimal neck muscles force across the different degrees of freedom.

Line 271-274. The authors make a good point of using Zheng’s model as the previous model was based on cadaveric data. However here it becomes clear that the muscles volume distribution is still based on the Vasavada’ model. Is there any limitation in relation to this approach?

Lines 309-327. One of the main limitations of Vasavada’s model is related to the neck muscles paths, as they are not anatomically realistic, especially considering the superficial muscles (traps). The authors are proposing a scaling factors mainly based on the geometrical position of different nodes, which can be rather different from the anatomical equivalent. Although I understand and appreciate the rigorous attempt to solve this problem, the error in the muscle paths and nodes position might be greater than a rough 3D scaling. Previous use of wrapping surfaces was attempted, and it might help to improve such scaling. Please comment on this.

Line 360-362. This should be included in the methods.

Reviewer #2: In review of "Neck Musculoskeletal Model Generation through Anthropometric Scaling" a paper seeking to accurately estimate neck loads with a scaling parameter. The topic is definitely worthy of study and the ability to scale biomechanical models to the individual is also a valuable contribution to the literature.

Intro: There is no need to start this paper with a description of neck pain. Readers ought to be aware of the applications of biomechanically model. I would rather like to see an introduction into the main issues this paper and work will address perhaps in the broader context of the literature. (ie how does this paper build on neck models and why is that important - focus on why what you have done is important and don't distract from this with a discussion on neck pain that is really part of the work which will follow the development of the model)

Finally, the intro stops abruptly - summarize the introduction rather then leaving the reader hanging on your last thoughts.

Methods:

More details about the AGM software would be helpful it seems this study heavily relies on this work which means is also is subject to the assumptions made in this work - what are those assumptions?

Is the female scaled to the 50th percentile and then those values extracted from the male data to match the size of the 50th percentile female? It is unclear when you use "male (female)" what exactly you are doing and mean by this. (Okay this comes at the end of the first section of the methods but is reference before it is fully described. Consider re-organizing). Also please explain what does gender specific differences mean?

Prove to the reader the optimization techniques are not simply making the math work and the model remains physiologically meaningful.

Results:

"2. Neck Strength optimized models

he original male model was too weak in flexion, lateral bending, and axial rotation

and too strong in extension. " provide the reader with some magnitude to quantify this amount!

"After optimization, the strength of the 50th percentile male was improved significantly for most directions. " quantify this amount for the reader.

Based on Table 4 - what model do you trust more and why? The values for some muscles (esp. flexion muscles) are very very different. stern_mast 86.1 N to 221.58 N. That seems like a major difference in prediction.

Also Rec Cap is a very small muscle and has a predicted force of 92.6 N compare this to Stern Mast originally predicted at 86.1 N.

Was the model tested only in a neutral posture?

Discussion: If muscles had via points and smaller moment arms - how would this effect your predicitions - would this make muscle flexion force even higher for example?

Line 476 : In (8), it was - rephrase

Overall:

Refrain from using "significantly" if you are not citing a specific statistical difference. ie females have significantly smaller necks.

Finally - limited information of the original method of force estimation is provided. More details on the methods of the original force estimation (prior to optimization) would be helpful - was this done?

6. PLOS authors have the option to publish the peer review history of their article (what does this mean?). If published, this will include your full peer review and any attached files.

Reviewer #1: Yes: Dario Cazzola

Reviewer #2: No

---

## [Author Response · Author response to Decision Letter 0]

18 Nov 2019

The authors would like to thank the reviewers for their very helpful comments and suggestions. Our response to the reviewers' comments and suggestions are detailed in the attached "Response To Reviewers" letter.

---

## [Decision Letter · Decision Letter 1]

2 Jan 2020

Neck Musculoskeletal Model Generation through Anthropometric Scaling

PONE-D-19-18973R1

Dear Dr. Zhou,

We are pleased to inform you that your manuscript has been judged scientifically suitable for publication and will be formally accepted for publication once it complies with all outstanding technical requirements.

With kind regards,

Bernadette Ann Murphy, PhD

Academic Editor

PLOS ONE

Additional Editor Comments (optional):

Reviewers' comments:

Reviewer's Responses to Questions

**Comments to the Author**

1. If the authors have adequately addressed your comments raised in a previous round of review and you feel that this manuscript is now acceptable for publication, you may indicate that here to bypass the “Comments to the Author” section, enter your conflict of interest statement in the “Confidential to Editor” section, and submit your "Accept" recommendation.

Reviewer #2: All comments have been addressed

2. Is the manuscript technically sound, and do the data support the conclusions?

Reviewer #2: Yes

3. Has the statistical analysis been performed appropriately and rigorously? 

Reviewer #2: Yes

4. Have the authors made all data underlying the findings in their manuscript fully available?

Reviewer #2: Yes

5. Is the manuscript presented in an intelligible fashion and written in standard English?

Reviewer #2: Yes

6. Review Comments to the Author

Reviewer #2: Thank you for your careful considerations of the review. Well done with the paper. I have no further comments and you have address my concerns.

7. PLOS authors have the option to publish the peer review history of their article (what does this mean?). If published, this will include your full peer review and any attached files.

Reviewer #2: No

---

## [Editor Report · Acceptance letter]

9 Jan 2020

PONE-D-19-18973R1 

Neck Musculoskeletal Model Generation through Anthropometric Scaling 

Dear Dr. Zhou:

I am pleased to inform you that your manuscript has been deemed suitable for publication in PLOS ONE. Congratulations! Your manuscript is now with our production department. 

With kind regards,

on behalf of

Dr. Bernadette Ann Murphy 

Academic Editor

PLOS ONE